# Bats at an Altitude above 2000 m on Pirin Mountain, Bulgaria

**DOI:** 10.3390/ani14010126

**Published:** 2023-12-29

**Authors:** Heliana Dundarova, Vasil V. Popov

**Affiliations:** Institute of Biodiversity and Ecosystem Research, Bulgarian Academy of Sciences, 1 Tsar Osvoboditel, 1000 Sofia, Bulgaria; vasilvpopov@gmail.com

**Keywords:** Chiroptera, underground sites, swarming, highlands

## Abstract

**Simple Summary:**

Little is known about bat diversity in mountains above 2000 m a. s. l. in Europe. Here, we report data from a study carried out in the high-alpine karst landscape of Pirin Mountain, Bulgaria at an altitude ranging from 2300 to 2600 m a. s. l. Twenty bat species were detected using three inventory methods. Furthermore, we describe the highest bat swarming site in Europe, located at an altitude of 2600 m a. s. l.

**Abstract:**

The study describes a pilot survey on bats in the highest areas of Pirin Mountain. The methods included examining subfossil bone remains, mist-netting, and recording echolocation calls. The study was conducted in August 2002 and 2013 and from 2019 to 2020. While in general, bat diversity tends to decrease with increasing altitude due to harsher environmental conditions, the present study, despite a short period, reveals high diversity. Twenty species, more than half of the Bulgarian bat fauna, were detected. The recording and analysis of vocal signatures proved to be the best way to inventory bat diversity. At least 13 species were detected by this method. *Vespertilio murinus* and *Tadarida teniotis* together make up more than 60% of all reliably determined echolocation sequences. Significant activity was found for *Myotis myotis/blythii*, *Plecotus auritus*, *Eptesicus serotinus*, and *E. nilssonii*. The registration of the latter species is of considerable faunistic interest. It was previously only known from a single specimen at one location in the country. The sex and age structure of the bat assemblage suggests that it is likely a swarming assemblage. The area is the highest swarming location in Europe. The results provide valuable information on bat ecology and behaviour, which can be used to inform management and protection efforts.

## 1. Introduction

Southwest Bulgaria is a diverse region with varying habitats due to its numerous mountain ranges. High-altitude areas receive more precipitation and have lower temperatures, while the lowlands are drier and warmer. Despite its potential for nature conservation, there is limited knowledge of animal communities, particularly bat assemblages. The demand for efficient conservation measures leads to the need for comprehensive regional faunistic studies. Most of these studies concern the lower regions of the country, where the species richness is greatest and where the most significant bat caves occur [1]. Data concerning the high regions of the mountains, notably those above the tree line, are few and in most cases are records of individual species [2,3,4,5,6,7,8].

Pirin Mountain is located between the valleys of the Mesta and Struma rivers and the western Rhodopes Mountains. The climate is continental–Mediterranean. The study area is alpine in nature, formed mainly by glacial activity during the Pleistocene. It is characterised by a great folding of the earth’s surface, steep slopes, sharp, bare rocky peaks and ridges, and the presence of glacial forms and deposits including cirques, trough valleys, various types of moraines, carlings, screes, scree plumes, and avalanche hives, etc. The cirques Bayuvi Dupki and Banski Suhodol (Figure 1) lie just below the main karst ridge of Northern Pirin Mt., where the highest and steepest peaks are: Vihren (2914 m), Kutelo (2907 m), Banski Suhodol (2884 m), Bayuvi Dupki (2820 m), Kamenititsa (2726 m), Razlog Suhodol (2640 m), Muratov vrah (2669 m), and Hvoinati vrah (2635 m). The average altitude of the area is 2300 m, with exposed karst as the dominant landform. Steep slopes, above 51°, are characteristic of the highest sections of the cirque walls. Due to the underlying karst’s drainage, surface water is absent, and the cirques are dry. Dozens of caves, abysses, niches, and crevices have formed in the area. The cold period, with prolonged average daytime temperatures below 0 °C exceeds 200 days. Generally, above an altitude of 2100 m, with average daily air temperatures below 10 °C, there is little to no active plant growth aside from a few specialised alpine species and cryptogams. Some of the limestone caverns are characterised by the presence of ice and snow plugs that do not melt every summer. The significant accumulation of snow masses through avalanches, sheer walls, shading, light marble cliffs with a high albedo, and karst forms, which dissipate rapidly melted water, favouring the formation of permanent névé and two glacierets below the contemporary snow line [9].

In recent decades, thanks to the development of molecular genetics, the existence of several cryptic bat species has been documented, many of which have sympatric or highly overlapping ranges [10,11,12,13,14,15,16,17]. This has raised new challenges for regional field studies, which are most often based on traditional morphological characteristics for species identification. Intensive research in this regard has allowed researchers to later identify previously overlooked peripheral morphological features that have then enabled them to distinguish morphologically similar species. In most cases, however, a reliable determination is only possible based on a complex of features. Some of these features are related to external morphology and are suitable for the determination of mist-netted specimens [18]. However, in many cases, some of the most useful characteristics can be found in the skull and dentition. These are especially useful when determining museum specimens. On the other hand, its application in regional studies is limited due to the need to kill specimens, which in most cases is undesirable from both the conservation and ethical points of view [19]. These circumstances greatly complicate the accumulation of new information on the distribution and ecology of cryptic species. An opportunity to overcome this difficulty is to collect material from bats that have died of natural causes. Their remains can often be found in caves as mummified specimens and bones—including well-preserved skulls, mandibles, and teeth—the study of which can help more precisely determine the species composition of regional bat communities.

Acoustic methods have become a popular tool for bat surveys due to their non-invasive nature and ability to identify bats based on their echolocation calls. Currently, specialised software is used to analyse calls and can automatically classify bat species based on call signatures. Several problems that lead to the misclassification of species have limited automatic identification programs. In general, programs cannot take into account atypical bat behaviour, approaching calls, cluttered environments, the presence of social calls, problems with the quality of recorded calls, multiple recorded bat individuals and (or) species, and trends across a series of call pulses in addition to deviations from the expected ringing structure due to natural overlaps in call metrics, pulse shape, and (or) sequence pattern that exist between some species. Thus, expert validation is often required for accurate results, especially for less-common or complex species [20,21,22,23].

This paper aims to present bat data in a karst landscape above the tree line of Pirin Mt. based on three inventory methods (analysis of bone remains, catches with mist nets and records of echolocation signals), including morphological and echolocation data to support the species determinations, and to analyse some structural parameters of the bat community. These results are relevant to ecological and environmental issues related to climate change and conservation decision-making. 

## 2. Materials and Methods

### 2.1. Study Sites

The present study was carried out at five underground (Figure 2) sites in 2013 and from 2019 to 2020. Site 1: Cave 33 (Golyamata yama) in cirque Bayuvi Dupki, denivelation: −33 m, N 41°48′19″ E 23°23′16″, 2250 m a. s. l.; Site 2: Cave 29 in cirque Bayuvi Dupki, denivelation: −80 m, N 41°48′07″ E 23°23′15″, 2346 m a. s. l.; Site 3: Cave 30 in cirque Banski Suhodol, denivelation: −120 m, N 41°47′28.8312″, E 23°23′34.9548″, 2600 m a. s. l., situated at the base of the Koteski Chal ridge. The relief is highly fragmented with steep slopes and high ridges with two glaciers; Site 4: Base camp in cirque Banski Suhodol, an area rich in caves and rock crevices overgrown with *Pinus mugo*; N 41°47′23.6400″, E 23°24′04.5360″, 2277 m a. s. l.; Site 5: Caves 9–11 in cirque Banski Suhodol, denivelation: −407 m, N 41°47′19.12″ E 23°23′40.16″, 2563 m a. s. l.

### 2.2. Skeletal Remains

Material was collected from the surfaces of the floors of two caves in cirque Bayuvi Dupki—Sites 1 and 2—on 24 August 2002 by B. Petrov and P. Beron. The preliminary species list based on material from Site 1 was published by Beron et al. [5]. For the most part, the material consists of well-preserved skulls. In some cases, they were accompanied by the lower jaws and postcranial bones, mainly those of the forearms. All skulls were from adult but not old individuals (teeth had almost no wear marks). These remains may belong to wintering bats that could not leave the caves as they remained blocked with snow long after the end of winter. The degree of preservation makes it possible to conclude that the material is practically modern, but its accumulation still probably spans a relatively long period. Thus, it can be considered a representative sample of the modern bat community in the area, especially those species reliant on caves during some parts of their life stages. Osteological material was deposited in the collections of the Institute of Biodiversity and Ecosystem Research at the Bulgarian Academy of Sciences. The material was examined with a binocular microscope and measurements taken with a dial calliper at an accuracy of 0.05 mm or with an ocular micrometre at an accuracy of 0.001 mm. Measurements included condylobasal length (CBL), zygomatic breadth (ZB), coronary lengths of the upper tooth rows (M1–M3, P4–M3, C–M3), interorbital breadth (IO), length of bulla ossea (LBO), length of the mandible (lmd), height of coronoid process (hpc), and height of the mandibular ramus under m^2^, measured lingually (hmd/m^2^). For more details, see Peshev et al. [24].

### 2.3. Capturing and Bat Identification

Bats were captured on 26 August 2013, 28–29 August 2019, 28–29 August 2020, 30 August 2020, and 1 September 2020 using a mist net with a length of 3 m and a mesh size of 16 mm at sites 3 and 4 in the Banski Suhodol cirque. The mist-netting period started 30–45 min before twilight and ended a few hours after midnight. Each bat was placed in a separate cloth bag. Species identification followed the field guide of Dietz and von Helversen [18]. For each individual, the main morphometric measurements—including the lengths of the forearm (FA), the fifth finger (D5), third finger (D3), and the upper tooth row (CM3)—were taken using plastic callipers. The epiphyseal ossification of the fingers of the thoracic limb and abrasion of the teeth were examined to age the captured specimens as adults or juveniles (born that summer) [25]. After identification, the bats were released at the site of their capture. The capture and handling of the animals were carried out with permissions No.” 948/03.06.2013, № 525/03.06.2013, № 2026/07.11.2018, № 760/15.11.2018, № NPP-97/05.02.2020 and № NPP-460/29.04.2021 from the Ministry of Environment and Water of Bulgaria and Pirin National Park. To test differences in the sex ratio, chi-square statistics, as implemented in Pearson’s chi-squared test for count data function (chi-sq.test), from the stats 3.6.2 R package https://cran.r-project.org/bin/windows/base/old/3.6.2/ (accessed on 12 November 2020) were used.

### 2.4. Acoustic Detection

Bats’ calls were recorded at Sites 3, 4 and 5 in the Banski Suhodol cirque using an M500-384 USB Ultrasound Microphone and the BatSound Touch recording program, Version 1.3.7, which saves the recordings as 16-bit wav files. At Site 3 (near Cave 30), recordings were collected on 27–28 August 2019 from 20:42 to 00:40 and on 30–31 August 2020 from 20:30 to 00:00. At Site 4, bat calls were recorded on 25 August 2019 from 20:49 to 22:05 and on 3–4 September 2020 from 20:30 to 00:00. At Site 5, recording was undertaken on 09/01–02/2020 from 20:30 to 00:00.

The recorded wav files were transferred to a computer using Kaleidoscope Pro 5.6.2 (Wildlife Acoustics Inc., Maynard, MA, USA). Through the batch option in bat analysis mode, the initial files from the recorder were divided into 1 s fragments, and those that did not contain bat calls according to the default signal detection parameters were deleted. Given the large volume of acoustic material, the length of 1 sec offers an optimal length in terms of subsequent manual verification, as it allows each sonogram to be fully opened on the computer screen, saving the need for scrolling and each call is well depicted with its “height” corresponding approximately to its length.

The program offers an automated species recognition, with varying degrees of accuracy, of the registered signals in each one-second audio file. The determination is based on measurements made on zero-crossing conversions of the recorded full-spectrum signals. As a result of this process, 6516 one-second bat call sequences with automatic identifications were obtained. The auto-identifications were considered as a starting point for subsequent manual vetting. Given the large number of sequences, to speed up and to facilitate the verification process, the proposals made by Kaleidoscope were compared with those made by BatScope 4. This program offers automatic detection based on 59 measurements for each full-spectrum signal. The generic support vector machine algorithm was used and provided 89% correct determinations [20]. BatScope 4 allows a large number of sonograms to be visualised simultaneously. This capability enabled us to quickly manually remove the numerous files containing only calls of the display song of *Vespertilio murinus*, identified by both programs as *Tadarida teniotis*. Nearly half of the sequences (3416) were excluded on this basis. In some cases, the social calls of *Myotis* spp., *Eptesicus serotinus*, and *Nyctalus noctula* [26] or fragments of *Vespertilio murinus* display songs were identified by both programs as *Plecotus* spp. Such sequences were also excluded from further analysis. After removing these files, the columns of the BatScope 4 “metadata.csv” file containing the automatic signal determinations in each of the remaining 2916 one-second sequences, representing search phase echolocation calls, were added to the Kaleidoscope’s analogous “id.csv” file. One-second sequences sorted by species using both programs (660) were subjected to a second round of more detailed verification in Kaleidoscope, with a special emphasis on the quality of the signals, especially the representativeness of the measurements made by the programs on which the proposed species identification was based. The final decision relied on generally accepted diagnostic characteristics of the signals [27,28,29] and comparative analysis with securely determined recordings available on the internet as well as in the reference databases of the programs used. This approach effectively handles substantial amounts of acoustic data within a practical timeframe. It ensures standardisation, transparency, objectivity, comparability, and reproducibility of the results. To generate more comprehensive insights, a sample of 179 sonograms representing all acoustically recorded species has been deposited in the ChiroVox database with the unique identification numbers A004538–A004717 (https://chirovox.org/country_map.php?q=Bulgaria) (accessed on 10 November 2023), allowing independent validation of taxonomic determinations.

## 3. Results

### 3.1. Skeletal Remains

Based on the subfossil material, seven species have been identified (Table 1). The determination of fragmentary skull materials in some bat groups, such as small species of the *Myotis* and *Plecotus* genera, is complicated and requires special reasoning. Some of the morphological features of the skulls available in these groups are specifically presented here to support their species’ identification.

Small *Myotis* of the *mystacinus* group. In the Balkan Peninsula, several morphologically similar species usually denoted as the “*M. mystacinus* group” coexist [11,12,30,31]. Among the remains studied, two skulls were assigned to *M. mystacinus* based on their size and peculiarities of the occlusal surface of the upper cheek teeth. The total length of these skulls (TScL = 13.2, 13.3 mm) is below the variability range of *M. brandtii* [30]. In M1 and M2, the paraloph is well developed and represents a high ridge; the paraconule is missing; the metaloph is very low. The anterolingual cingular tubercle of P4 is well developed, but relatively low (Figure 3). The small size (Table 1), the lack of paraconulus, and the poorly developed metaloph allow one to exclude these two skulls as belonging to *M. brandtii*. In skull size, they are below the lower variability range of *M. aurascens* [30], which was synonymized with of *M. davidii* [32]. They are also distinguished from this species by the presence of the anterolingual cingular tubercle of P4 (Figure 3). The condylobasal length of these two skulls is above the upper limit of the variability of the type series of *M. alcathoe* from Greece [11]. The material studied is also separated from this species by the absence of paraconules on the upper molars. Thus, due to its peculiarities, this material corresponds completely to *M. mystacinus* as described by Benda and Tsytsulina [30]. 

One skull in this morphological group is relatively large (Table 1), falling within the upper margin of the variability of *M. brandtii* [30]. M1 and M2 show well-developed paraconules and para- and metalophs. The anterolingual cingular tubercle of P4 is very high (Figure 3). These features prove this skull to be one of *M. brandtii* in the context of the description presented by Benda and Tsytsulina [30].

*Plecotus*. In the neighbouring countries of Bulgaria, in addition to the two widespread European species, *Plecotus auritus* and *P. austriacus*, two more species, *Plecotus kolombatovici* and *P. macrobullaris*, also occur [12,15,33]. The first two species differ greatly in skull size; *P. austriacus* is larger. Regarding the condylobasal lengths of the skulls, there is virtually no overlap between them [7]. The other two species are intermediate in size and show overlap in most cranial sizes. Thus, identification based on skulls only is difficult. The distribution of *P. kolombatovici* is strongly associated with warm summer temperatures and relatively high winter precipitations, typical of the Mediterranean climate [34], and its occurrence in the high regions of Pirin Mt. is unlikely. In contrast, *P. macrobullaris* is a species of alpine distribution, occurring in the high mountains of southern Europe. Its presence in the high mountains of Bulgaria is possible but not yet established [1,35]. Discriminant functions based on skull measurements have been proposed to distinguish *P. macrobullaris* from *P. auritus,* e.g., Spitzenberger et al., and Pavlinić and Đaković [36,37]. However, their application to fragmented material is not possible. Simplified options have been proposed, with particular reference to the use of the tympanic bullae length and the coronary length of the upper dentition, C-M3 [7,38,39]. However, based on genetically determined individuals from Western Europe, it has been shown that in these measurements, there is also overlap between these species [4]. One specimen of the study material retains the tympanic bullae, which allows its dimensions (Table 1) to be compared with the data of Benda and Ivanova [7] and Andriollo and Ruedi [40]. It falls within the variability of *P. auritus* but below the overlapping range of *P. auritus* and *P. macrobullaris*. In terms of other measurements, the sample of Pirin skulls is homogeneous. Therefore, the analysis of the subfossil material corresponds to the identified mist-netted individuals and indicates the presence of *P. auritus* only in the study area. 

### 3.2. Mist Netting

A total of 60 individuals belonging to 6 species were captured. The majority of captured animals were subadults. The most abundant species was *Plecotus auritus*, followed by *Myotis nattereri*, *M. bechsteinii*, and *M. blythii*. Two species, *Myotis emarginatus* and *M. mystacinus*, were represented by a few specimens each. Most of the captured bats were males (78.6%), and the sex ratio was significantly different from the expected male:female ratio of 1:1 (X-squared = 9.1429, df = 1, *p*-value = 0.002497), see Figure 4. However, in the dominant species, *P. auritus*, although males dominated (7 vs. 4) the sex proportions did not differ statistically from 1:1 (X-squared = 0.81818, df = 1, *p*-value = 0.3657). Excluding this species, male bias remained statistically significant (X-squared = 9.9412, df = 1, *p*-value = 0.001616). This is not surprising given that for three species, no females were captured.

### 3.3. Acoustic Detecttion

A large selection of the signals in the studied material belong to a group of bats emitting calls consisting of a frequency-modulated pulse, followed by a quasi-constant frequency pulse in the interval 20–30 kHz. This sonotype includes the species *Vespertilio murinus*, *Nyctalus leisleri*, *Eptesicus serotinus*, and *Eptesicus nilssonii*. In this group, the last species is of special interest. This northern species has been detected with certainty in Bulgaria only once, in the high parts of the Rila Mountains [2]. It can be assumed that it probably also occurs in the other high mountains. Distinguishing the species belonging to this acoustic group is difficult, as there is considerable overlap in most quantitative call characteristics. The presence of *E. nilssonii,* is recorded by both programs (Kaleidoscope and BatScope 4) and is always second in terms of relative share compared to other species in the group. This fact is noteworthy considering that the share of each of the other species is different for the two programs. The majority of signals from this acoustic group were assigned to *N. leisleri* by Kaleidoscope and to *V. murinus* by BatScope 4. These conflicting results are not surprising given the wide frequency range overlap of calls for these two species in the 22–25 kHz range [24,41]. In this context, the differences are most probably due to the different samples on which the programs’ algorithms were trained. It can be said that the results of BatScope 4 regarding this acoustic group are more accurate as long as the detected dominance of *V. murinus* corresponds to the high proportion of signals representing the display song of this species. The accuracy of the results of this program is most likely determined by the fact that the models are trained on large samples of species from geographically nearby and ecologically similar areas [21]. In comparative analyses [22,42], it was discovered that *V. murinus* and *N. leisleri* were identified with little to no accuracy by Kaleidoscope, whereas it showed a relatively high level of accuracy for *E. nilssonii*. In cases of misidentification, *E. nilssonii* was most commonly determined as *N. leisleri*. These results show that *E. nilssonii* can be identified with a high degree of accuracy thanks to the fact that its upper frequency range of 28–32 kHz falls outside the overlapping range of the other two species [41]. Based on these points, we can conclude that the records of *E. nilssonii* are accurate and demonstrate its presence in the study area. However, due to the similarity of the signals and the wide overlap in their diagnostic characteristics as well as the lack of non-acoustic evidence for the presence of these species in the area, an open nomenclature was used for these species, with the exception of *Vespertilio murinus*, whose presence in the area is evidenced by the numerous social sounds (Table 2).

*Myotis*. In this group, the following species were identified using Kaleidoscope and BatScope 4: *Myotis emarginatus*, *Myotis mystacinus*, *Myotis brandtii*, *Myotis daubentonii*, *Myotis nattereri*, *Myotis bechsteinii*, *Myotis myotis*, and *Myotis blythii*. The verification of signals from this group is mandatory, as they are generally weak and often misidentified by automatic identification programs. The identification of *M. emarginatus* was not in doubt (https://chirovox.org/country_map.php?q=Bulgaria) (accessed on 11 November 2023) (A004597, A004598). According to a comparative study by Rydell et al. [22], Kaleidoscope gives a very high rate of correct identification for *Myotis daubentonii* (88.2%). Manual determination of the present material confirms this. Through the verification process, it has been found that the automated identification of *Myotis* calls primarily reflects signal quality rather than species variations. Consequently, the signals of *Myotis bechsteinii*, *M. nattereri*, *M. brandtii*, and *M. mystacinus* were not assessed independently and were classified as *Myotis* spp.

*Plecotus* spp. Calls from these species are weak and often misidentified by automatic detection programs. Only signals identified by both programs were assigned to the species of this group. 

*Tadarida teniotis—Nyctalus lasiopterus*. These two species emit signals in the low-frequency range and are often difficult to distinguish. In general, the second species has a slightly higher frequency, and the signals are steeper. The presence of both types of signals in sequences that apparently belong to different individuals indicates that both species are present in the area. The typical calls of *Tadarida teniotis* in free flight predominate in the recordings (Table 2). According to the data, it appears that *T. teniotis* is the most prevalent species in the region, a fact that has been substantiated by sightings during twilight hours. The terrain characteristics, including sharp ridges and rocky cliffs with abundant crevices and cavities, make an especially favourable habitat for this species.

*Rhinolophus hipposideros*. The average value of the peak frequency of the available signals is 105.983 kHz (ChiroVox ID—AA004648, AA00464). This value, although close, is above the upper limit of overlap with *Rh. euryale* and is close to the average value of *R. hipposideros* [43].

Overall, the analysis of acoustic data detected at least 13 bat species (Table 2). The highest call activity was found for *Vespertilio murinus*, represented by 3552 one-second call sequences (search and display calls). Other frequently detected species or groups were *Tadarida teniotis, Eptesicus nilssonii, Eptesicus serotinus, Nyctalus leisleri, Plecotus* spp., and *Myotis myotis/blythii*. For all other species or species groups, only a few recordings were obtained. 

Seven species were represented by more than 10 sequences, allowing us to follow the dynamics of activity during the first half of the night (Figure 5). *Eptesicus serotinus*, *E. nilssonii*, and *Vespertilio murinus* were most active from 21–22 h. *T. teniotis* was most active just after dark, and *N. leisleri* had the highest activity at the beginning of the night. *Myotis myotis/blythii* and *Plecotus auritus* were most active around midnight.

## 4. Discussion

In total, 20 bat species were recorded during the study. In total, 7 species were identified from skeletal material, 6 via mist-netting, and at least 13 acoustically. Compared to the results of the other methods, the last was the most effective. The presence of virtually all species detected by the first two methods was confirmed using vocal signatures (Table 2). The results confirm the advantages of acoustic data for studying bat communities in that short-term acoustic surveys can nearly equal or exceed the known species richness of bat communities and can add species that are difficult to detect using conventional capture methods [44,45]. 

In Bulgaria, bat communities at the highest altitudes are poorly studied. So far, seven species have been recorded at elevations above 2000 m a. s. l. There is a report from Rila Mt., where an *Eptesicus nilssonii* carcass was found in the Ribni Ezera hut at 2250 m a. s. l. [2]. Sporadic data from the alpine zone of Pirin Mt. are provided by Beron [3,4,5,6], Benda and Ivanova [7], and Benda et al. [8] of *Myotis blythii*,* Pipistrellus nathusii,* and *Plecotus auritus* from Vihrenska propast at 2500 m a. s. l.; *Vespertilio murinus* from Nishata cave at 2000 m a. s. l. and Koncheto shelter at 2760 m a. s. l.; and of *Myotis daubentonii* and *M. nattereri* from Cave 29 (Site 2). These species (excluding *P. nathusii*) were confirmed in the present study. In the lower part of Pirin Mt., in the range 1000–2000 m, almost the same number of species (n = 21) as in the current short-term study have been recorded over nearly a century of studies [3,8,46,47,48,49,50,51,52]. Differences include the absence of *Plecotus austriacus*, *Rhinolophus euryale*, *R. ferrumequinum*, and *Myotis aurascens* from the study area. Most of these species are thermophilic and their absence above 2000 m is understandable. 

In a similar study in the Alps from 2012 to 2016 between the altitudes of 2250 m and 2761 m, 12 species were recorded using ultrasound recording and mist netting [53,54]. Most of them were also present in our study area, except for *Pipistrellus nathusii* and *P. pygmaeus*. With longer studies, these species would probably also be found on Pirin above 2000 m, since the first species is known from the lower parts of the mountain and the second is a frequent species in the mountains and northern regions of Bulgaria according to echolocation data (Popov, unpublished data). However, in addition to the alpine species, the current study also recorded *Rhiolophus hipposideros* (Table 2), an unusual occurrence above 2000 m a. s. l. The presence of *R. hipposideros* at this elevation could be a positive correlation between the warm thermals and food availability at this time of year in the alpine zone of the mountain. The higher species diversity found in Pirin Mt. is likely due to its more southerly location and higher average temperatures compared to the Alps. 

The results of the mist-netting reveal interesting peculiarity of the bat community, namely the disproportion in the sex ratio in favour of males. As far as the sex ratio is concerned, similar results were observed in the high-altitude areas of the Central European and Apennine Mountains [54,55,56,57,58]. According to some studies, the low abundance or absence of females at higher altitudes in summer is a common phenomenon and is related to energetic requirements during the pregnancy and lactation periods [59,60,61]. This tendency is most evident in species whose optimum decreases with altitude. Conversely, for species that prefer higher altitudes, this has not been demonstrated [59]. This pattern is to some extent confirmed by our data. *Plecotus auritus*, which in Bulgaria is a strict mountain-dweller [1], is represented by the largest number of females. 

According to some authors [62,63], in contrast to the proportion of females, the proportion of males and subadults of both sexes increases with altitude. This could be explained by their ability to conserve energy during torpor when conditions are not good. However, according to other studies, in some species the proportion of adult individuals increases with altitude. Juveniles are believed to remain at lower altitudes to benefit from greater nutritional resources (similar to adult females) and to maintain contact with the mother (e.g., [59,60,61]). A similar age-dependent elevational trend has been observed during the swarming period in the Carpathians [58]. 

The peculiarities of the sex–age structure of the bat assemblage, as well as the season during which they were registered (late August), indicate that it most likely represents a swarming aggregation. The registered territorial/social calls of *V. murinus* are also possible evidence of swarming behaviour. The prime task of swarming is to localise mates and transitional stops during migration and/or to check suitable hibernacula [64,65,66,67,68,69,70,71,72]. During swarming, bats from various summer roosts aggregate [73,74] and swarming sites act as intermediate stopovers during migrations [70]. Swarming is supposedly important for introducing subadults to suitable caves for hibernation [62,64,75,76], which can explain the dominance of young individuals since most of the potential hibernacula are concentrated within the study area. Skeletal remains presented in this study prove that karst cavities are used for hibernation by at least seven bat species.

A male bias in swarming bats is typically expected [64,69,74,77,78]. We registered male bias activity at the entrance of Cave 30 (Site 3) at 2600 m a. s. l. Swarming sites between 1294 and 1907 m a. s. l. in the Carpathian Mountains [58,75] and between 1828 and 2050 m a. s. l. in the Italian western Alps [79] have been recorded to date. Most studies of activity in the European mountains up to 2500 m a. s. l. level are based on acoustic records [51,52,53]. Thus, Cave 30 (Site 3) is the highest currently known swarming location in Europe.

Considering the location of Pirin Mt. in southwestern Bulgaria between the Rila and Rhodopes Mountains, Cave 30 (Site 3) is a possible transient stopover point for bat migration between the summer and winter roosts. In all reported species, vertical migrations between their day roosts and the swarming sites were observed [65,72,80]. Previous studies have demonstrated that *Myotis* cf. *mystacinus*, *M. daubentonii*, *Vespertilio murinus*, *Eptesicus nilssonii*, *Pipistrellus nathusii/kuhlii*, *P. pipistrellus*, *P. pygmaeus*, *Nyctalus noctula*, *N. leisleri*, *Barbastella barbastellus*, and *Plecotus auritus* cross the Alps up to 2500 m a. s. l. [51,52], while species such as *V. murinus, P. nathusii, N. noctula*, and *N*. *leisleri* are long-distance migrants [81]. 

The results of the presented faunistic survey play a pivotal role in monitoring biodiversity changes within the broader context of global environmental changes, such as climate change, habitat degradation, and species invasions. In particular, they establish baseline data on species diversity, distribution, and abundance within a poorly studied region. The obtained data serves as a reference point to monitor changes over time. Continuous monitoring through faunistic surveys allows for adaptive management strategies. As changes are detected, conservation strategies can be adjusted to address emerging threats or protect vulnerable species.

## 5. Conclusions

Current research has shown that the karst landscape in the high parts of Pirin Mountain, despite harsh climate conditions, provides swarming sites which can be used as transitional stops and probably hibernacula for bats.

## Figures and Tables

**Figure 1 animals-14-00126-f001:**
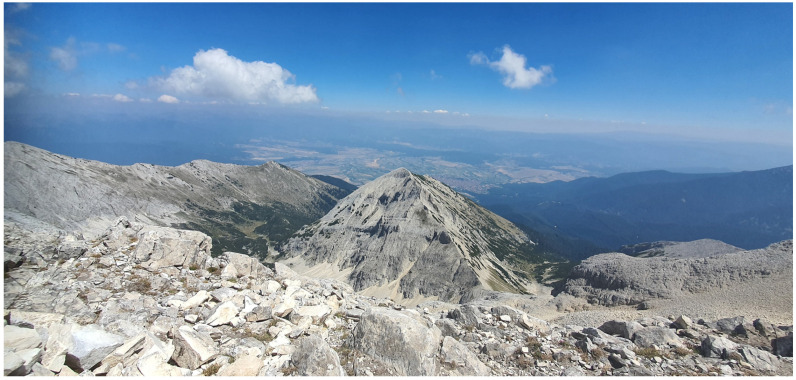
The main karst ridge of the Northern Pirin Mountain.

**Figure 2 animals-14-00126-f002:**
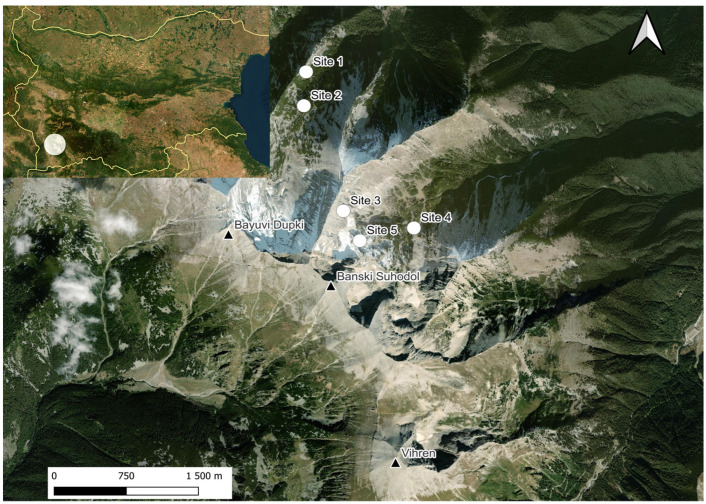
Map of studied localities in Pirin Mt. Site 1 marks the location of Cave 33, Site 2—Cave 29, Site 3—Cave 30, Site 4—the “Base camp”, and Site 5—Caves 9–11. Triangles indicate mountain peaks.

**Figure 3 animals-14-00126-f003:**
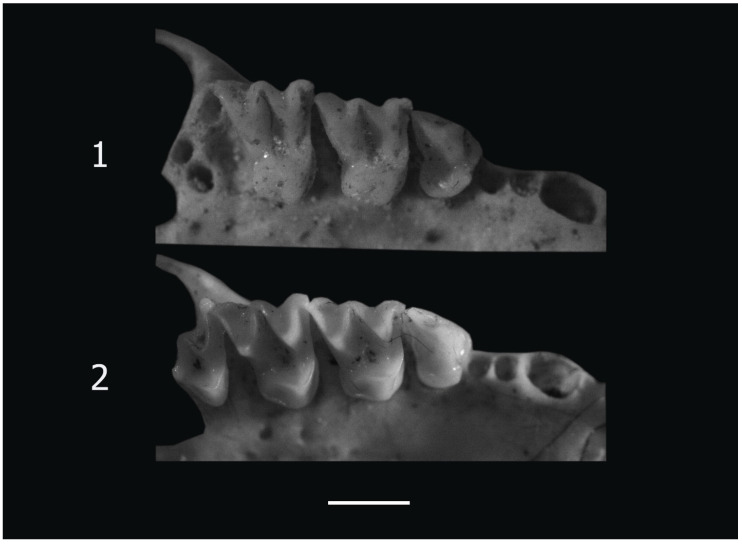
Right upper tooth rows of *Myotis* species from Site 1 (Cave 33, Bayuvi Dupki), Pirin. **1**. P4–M2 of *Myotis brandtii*, **2**. P4–M3 of *M. mystacinus*. The white line is a scale bar represents 1 mm.

**Figure 4 animals-14-00126-f004:**
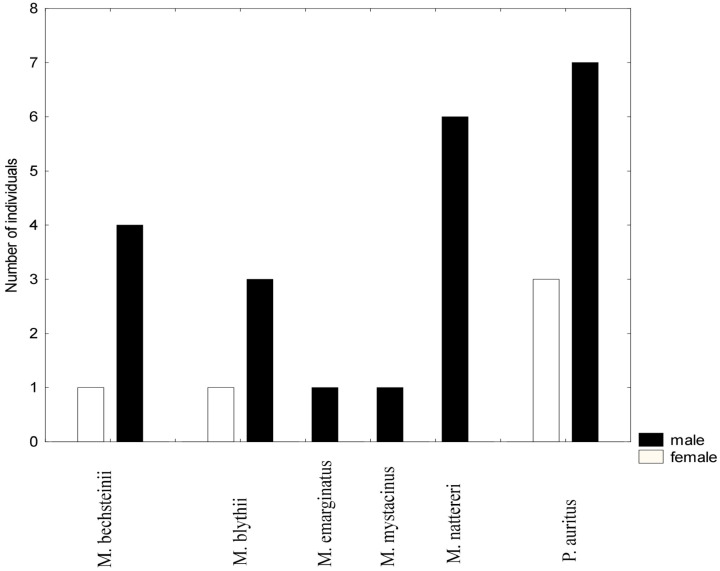
Bar plot showing the sex ratio in bat species caught using mist nets, Site 3 (Cave 30, 2600 m a. s. l.).

**Figure 5 animals-14-00126-f005:**
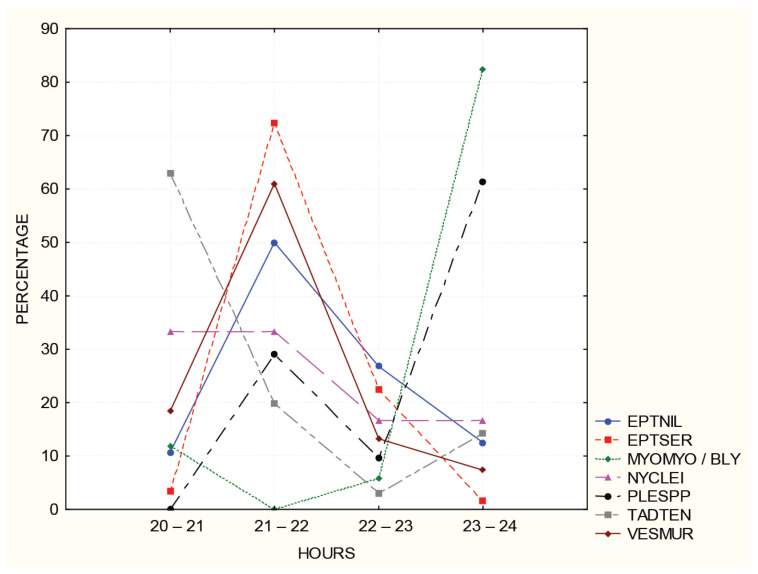
The temporal activity patterns of bats represented by more than 10 one-second sound recordings. Sunset at 20:40.

**Table 1 animals-14-00126-t001:** Cranial, mandibular, and dental measurements of bats collected from Cave 33 (Site 1) and Cave 29 (Site 2) in cirque Bayuvi Dupki, Pirin Mt. Measurements are presented in mm as mean, number of specimens (N), minimum (Min) and maximum (Max), and standard deviation (SD). For samples of two or less, no descriptive statistics are presented.

Species	Measurements	N	Min	Mean	Max	SD
*Myotis myotis*	lmd	1	-	18.1	-	-
	hpc	1	-	6.5	-	-
	hmd/m2	1	-	2.6	-	-
	i-m3	1	-	11.8	-	-
*Myotis blythii*	CBL	6	20	20.53	21.3	0.520
	C-M3	2	9	-	9.2	-
	P4-M3	6	6.3	6.58	6.7	0.147
	M1-M3	6	5.2	5.35	5.5	0.138
	zg	6	13.6	14	14.4	0.283
	io	6	5.2	5.28	5.5	0.133
	lmd	3	15.8	16.2	16.3	0.265
	hpc	3	5.2	5.2	5.5	0.173
	hmd/m2	3	2.1	2.6	2.6	0.252
	i-m3	1	-	10.8	-	-
	p4-m3	1	-	7.1	-	-
	m1-m3	1	-	5.7	-	-
*Myotis bechsteinii*	CBL	2	16.2	-	16.6	-
	P4-M3	2	4.8	-	4.9	-
	M1-M3	2	3.8	-	3.95	-
	ZB	1	-	10.5	-	-
	io	1	-	4.4	-	-
*Myotis nattereri*	CBL	3	14.4	14.58	14.8	-
	P4-M3	3	4.4	4.5	4.6	-
	M1-M3	3	3.5	3.6	3.7	-
	ZB	2	9.7	9.8	9.9	-
	io	3	3.6	3.9	4.2	-
*Myotis mystacinus*	CBL	2	12.6	-	12.7	-
	P4-M3	2	3.7	-	3.7	-
	M1-M3	1	-	3.01	-	-
	ZB	2	8	-	8	-
	IO	2	3.5	-	3.6	-
	lmd	1	-	9.7	-	-
	hpc	1	-	2.7	-	-
	hmd/m2	1	-	1.1	-	-
	i-m3	1	-	6.4	-	-
*Myotis brandtii*	CBL	1	-	13.6	-	-
	P4-M3	1	-	4.1	-	-
	M1-M3	1	-	3.3	-	-
	ZB	1	-	7.9	-	-
	IO	1	-	3.7	-	-
*Plecotus auritus*	CBL	10	14.7	15.19	15.5	0.260
	I1_M3	2	6.1	-	6.4	-
	C-M3	2	5.3	-	5.5	-
	P4-M3	10	4.1	4.25	4.4	0.097
	M1-M3	10	3.2	3.34	3.6	0.107
	ZB	9	8.4	8.6	9	0.194
	IO	11	3.5	3.60	3.7	0.072
	LBO	1	-	4.1	-	-
	lmd	1	-	10.6	-	-
	hpc	1	-	3.4	-	-
	hmd/m2	1	-	1.4	-	-
	i-m3	1	-	6.9	-	-

**Table 2 animals-14-00126-t002:** List of bat species registered in the study area.

Bat Species	Bone Remains	Mist Net	Ultrasound Sequences
*Rhinolophus hipposideros*	-	-	1
*Barbastella barbastellus*	-	-	10
*Myotis myotis*	1	-	-
*Myotis blythii*	9	12	
*Myotis myotis/blythii*			23
*Myotis emarginatus*	-	4	6
*Myotis bechsteinii*	2	8	-
*Myotis nattereri*	3	12	-
*Myotis brandtii*	1	-	-
*Myotis mystacinus*	2	3	-
*Myotis daubentonii*	-	-	5
*Myotis* spp.	-	-	24
*Hypsugo savii*	-	-	4
*Pipistrellus pipistrellus*	-	-	3
*cf. Eptesicus serotinus*	-	-	58
*cf. Eptesicus nilssonii*	-	-	56
*Vespertilio murinus*	-	-	136
*Nyctalus lasiopterus*	-	-	14
*cf. Nyctalus leisleri*	-	-	6
*Nyctalus noctula*	-	-	3
*Plecotus auritus*	12	21	-
*Plecotus* sp.	-	-	39
*Tadarida teniotis*	-	-	272

## Data Availability

The authors can provide the data if needed.

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
