# Peer review of "Bats at an Altitude above 2000 m on Pirin Mountain, Bulgaria"

_animals, 2023, doi:10.3390/ani14010126_

Round 1

Reviewer 1 Report

Comments and Suggestions for Authors

Dear editor and authors,

Thank you for the opportunity to review this paper. I find this paper well written and interesting. It is mainly a descriptive study, based on a survey but still important to publish. The scientific style in writing is good. I suggest minor revision and I have just some small comments.

1. Annex

I do not find the annex very useful. It contains just some photos of the study sites and sonogram from recordings. However, the sonograms are difficult to evaluate since the scale is very small and it is impossible to see the numbers on the x- and y-axis. Sonograms are not needed.

2. The connection between introduction and conclusions.

In the introduction it is written ”These results are relevant to ecological and environmental issues related to climate change and conservation decision-making.” However, nothing is mentioned about that in the discussion and I wonder how is this relevant for climate change and conservation decisions. Please, add something about that in the discussions.

3. Spelling.

Line 106: “2600 a.s.l.m” please, change to: 2600 m a.s.l.”

Line 110-111: “2563 m.” please add “a.s.l.”

Line 226: “Table 2” should be table 1”

Line 310: “Myotis nattereri” should be in italic.

Line 314: “Reydel et al.” should be “Rydell et al.”

Table 3: wrong spelling of Myotis daubentonii and Nyctalus noctula

4. Table 3 and 4.

Table 3 show a total list of the bats identified in the study. Table 2 shows the recorded bats. I don’t understand why the numbers in the column “ultrasound sequences” in table 3 not is the same as in table 2? It seems that table 2 only is a subsample of all recordings, why? In table 2, M. blythii and M. Myotis are combined to M. Myotis/blythi, and the species within the genus Plecotus are combined to Plecotus spp. However, in table 2, in the column “ultrasound sequences”, there are no Myotis blythii, but only M. Myotis, and the Plecotus is identified to P. auritus. M. bechsteinii is missing in table 2, but in table 3 there are 24 recordings of M. bechsteinii. Why is not M. bechsteinii included in table 2?

Table 2 is very ambitious and contains much more information than needed. I do not understand “Fc = characteristic frequency of call pulses”. However, I guess it is the same as Frequency at maximum energy, FME, which I think is a more correct description. In general, I don’t find mean values very useful for identification. The range is more important, for example the range of FME in relation to pulse repetition rate. I suggest that the authors should consider if table 2 is needed.

Why is N. noctula not included in table 2?

5 Discussion

Very interesting discussion. However, I would like to see a better connection to the promises in the introduction. The dominance of male is interesting, however, the data set is small. Why not increasing the data set by using DNA analysis of the skull? The result demonstrates very well that the caves in the mountains are used for hibernation and swarming. What about the rest of the year, are there any bats in the mountains during summer? Are there colonies in the mountains? In the conclusion it is written: Mountain, despite harsh climate conditions, provides suitable summer roost and hibernacula, as well as swarming places.” However, I can’t see any results in this paper supporting occurrence of summer roosts.

Finally, I just want to congratulate the authors for a very nice and well written paper.

Author Response

Rev 1

Thank you for the opportunity to review this paper. I find this paper well written and interesting. It is mainly a descriptive study, based on a survey but still important to publish. The scientific style in writing is good. I suggest minor revision and I have just some small comments.

  1. Annex

I do not find the annex very useful. It contains just some photos of the study sites and sonogram from recordings. However, the sonograms are difficult to evaluate since the scale is very small and it is impossible to see the numbers on the x- and y-axis. Sonograms are not needed.

Response: We agree with this note. In the context of the suggestion of one of the reviewers, a representative sample of sonograms for each identified species is (proposed) deposited in a ChiroVox – 179 wav-files, accompanied by a metadata file containing detailed information about each sonogram, as required by the database.

  1. The connection between introduction and conclusions.

In the introduction it is written” These results are relevant to ecological and environmental issues related to climate change and conservation decision-making.” However, nothing is mentioned about that in the discussion, and I wonder how is this relevant for climate change and conservation decisions. Please, add something about that in the discussions.

  1. Spelling.

Line 106: “2600 a.s.l.m” please, change to: 2600 m a.s.l.”

Response – corrected

Line 110-111: “2563 m.” please add “a.s.l.”

Response – corrected

Line 226: “Table 2” should be table 1”

Response: The table numbers have been completely revised in the context of dropping Table 2.

Line 310: “Myotis nattereri” should be in italic.

Response – corrected

Line 314: “Reydel et al.” should be “Rydell et al.”

Response – corrected

Table 3: wrong spelling of Myotis daubentonii and Nyctalus noctula

  1. Table 3 and 4.

Table 3 show a total list of the bats identified in the study. Table 2 shows the recorded bats. I don’t understand why the numbers in the column “ultrasound sequences” in table 3 not is the same as in table 2? It seems that table 2 only is a subsample of all recordings, why? In table 2, M. blythii and M. Myotis are combined to M. Myotis/blythi, and the species within the genus Plecotus are combined to Plecotus spp. However, in table 2, in the column “ultrasound sequences”, there are no Myotis blythii, but only M. Myotis, and the Plecotus is identified to P. auritus. M. bechsteinii is missing in table 2, but in table 3 there are 24 recordings of M. bechsteinii. Why is not M. bechsteinii included in table 2?

Table 2 is very ambitious and contains much more information than needed. I do not understand “Fc = characteristic frequency of call pulses”. However, I guess it is the same as Frequency at maximum energy, FME, which I think is a more correct description. In general, I don’t find mean values very useful for identification. The range is more important, for example the range of FME in relation to pulse repetition rate. I suggest that the authors should consider if table 2 is needed.

Why is N. noctula not included in table 2?

Response: At the suggestion of one of the reviewers, a representative sample of acoustic recordings of each species was uploaded in a public database - ChiroVox. This allows for a critical assessment of the determinations made and makes Table 2 redundant. It has been removed from the manuscript. However, here we will give some clarifications related to discrepancies in the data in tables 2 and 3 noted by the reviewer. The measurements that were presented in the table were automatically generated by Kaleidoscope for each sonogram. Since in many of the sonograms, there are signals of more than one species, and the program generates averaged values, they have no taxonomic value. The measurements in Table 2 were based only on those sonograms in which the signal measurements characterized only one species - hence the discrepancies in the figures in the two tables. N. noctula and other species were not included because being rare they were always accompanied by other species, so the generated measurements were not relevant. Since the measurements generated by Kaleidoscope are based on zero-crossing signals and not on the full spectrum, it does not measure the frequency of maximum energy. The characteristic frequency is a similar but not identical parameter. Moreover, being related to the shape of the signal, it is more stable, resp. more suitable for distinguishing acoustically similar species.

In addition to the above circumstances, some shortcomings in the layout of Table 3 also contributed to the ambiguities in the numerical data for the acoustically identified species. Corresponding corrections have been made.

5 Discussion

Very interesting discussion. However, I would like to see a better connection to the promises in the introduction. The dominance of male is interesting, however, the data set is small. Why not increasing the data set by using DNA analysis of the skull? The result demonstrates very well that the caves in the mountains are used for hibernation and swarming. What about the rest of the year, are there any bats in the mountains during summer? Are there colonies in the mountains? In the conclusion it is written: Mountain, despite harsh climate conditions, provides suitable summer roost and hibernacula, as well as swarming places.” However, I can’t see any results in this paper supporting occurrence of summer roosts.

Response: Relevant text at the end of the Discussion has been added, which makes the necessary connection with the intentions expressed in the Introduction.

The field surveys were done in late summer and thus provide insight into the species that inhabit this landscape during that season. In addition, the text states that the karst forms and cavities in the area provide conditions for the habitat of T. teniotis, which is one of the most abundant species in the area.

Finally, I just want to congratulate the authors for a very nice and well written paper.

Reviewer 2 Report

Comments and Suggestions for Authors

In this manuscript, the authors investigate bat diversity at high elevations in Bulgaria using various methods. This topic is very interesting, but I am concerned about analysis methods and interpretation of their results.
I will provide some comments along with references that might help improving the quality of the manuscript. 

DISCREPANCIES

From the sentence in the abstract (line 14) it is not clear that in all three years sampling was performed only at the end of August. In the abstract authors refer to data collected in 2013 and 2019-2020, and there is skeletal material included that was collected in 2002 (lines 116-117).

Lines 22-23 authors state „Within all species adult males and juveniles predominated.“ while in lines 259-260 they state „all the animals captured were juveniles“ (contradiction between all juveniles and adults+juveniles)

Lines 121-122: Please provide a reference to support this statement (that bats remained blocked due to snow)

Lines 219, 369 why do authors use Myotis aurascens? Based on Çoraman et al 2020 and EUROBATS resolution 8.2, M. aurascens (M. mystacinus bulgaricus) is synonym of M. davidii

Lines 431-433 (conclusios): It cannot be concluded that high parts of the Pirin Mountains provide suitable summer roosts without actual sampling during summer period and confirming reproduction (=existence of maternity roosts) there. Bats that were born that year are maybe just dispersing, it doesnt mean that they were born in that exact area

References:

Çoraman E, Dundarova H, Dietz C, Mayer F. Patterns of mtDNA introgression suggest population replacement in Palaearctic whiskered bat species. Royal Society Open Science. 2020 Jun 3;7(6):191805.

https://www.eurobats.org/sites/default/files/documents/pdf/Meeting_of_Parties/MoP8.Resolution%208.2%20Amendment%20of%20the%20Annex%20to%20the%20Agreement_0.pdf

ACOUSTIC IDENTIFICATION

Authors should be more cautious and conservative with acoustic identification. It is true that bat surveys in such habitats are difficult, and that acoustics represents a great tool for fauna surveys, but a more robust approach should be considered. It is unclear how recordings were made – was the device set up for continuous recording, or only when there was a bat flying? In case of later, was the recording ongoing as long as bat echolocation calls were detected?

Could authors justify why they chose to cut their recordings in 1s? There are some papers suggesting that 5s files are of good length to compromise the detection distance (Millon et al 2015). Did authors check noise files as well, to be sure there were no files rejected? I do not have experience with BatScope 4, but in my experience, Kaleidoscope sometimes discards files with horseshoe echolocation calls as “noise” files.

The next issue I would like to raise is identification to species level instead of sono-group for some „tricky“ species. There has been a lot of discussion about the unambiguous identification of certain species based on echolocation calls only (and in the absence of the social calls) such as species from genera Eptesicus / Nyctalus / Vespertilio. In many papers, authors group them into sono-group instead of identifying them to species level with lower certainty. In this paper, authors recorded display calls of V. murinus, which allows its reliable identification, but I would suggest authors to be a little more prudent with other species from the sono-group. For example, Rydell et al (2016) stress the very high percentage of erroneously identified recordings of species from this sono-group. The same authors point out frequent misclassifications of B. barbastellus. Authors in current ms present measurements of echolocation calls of this species. However, B. barbastella has two types of calls, and the authors did not specify which measurements (from which call type) were presented in their manuscript?

R. hipposideros presence was confirmed based only on a single pass, and call measurements were not provided in Table 2. R. hipposideros echolocation calls are known to overlap with ones from R. euryale (Győrössy et al 2020, Schofield et al 2022)

 References: 

Millon L, Julien JF, Julliard R, Kerbiriou C. Bat activity in intensively farmed landscapes with wind turbines and offset measures. Ecological Engineering. 2015 Feb 1;75:250-7.

Rydell J, Nyman S, Eklöf J, Jones G, Russo D. Testing the performances of automated identification of bat echolocation calls: A request for prudence. Ecological indicators. 2017 Jul 1;78:416-20.

Győrössy D, Győrössy K, Estok P. Comparative analysis of the echolocation calls of the lesser horseshoe bat (Rhinolophus hipposideros) and the Mediterranean horseshoe bat (Rhinolophus euryale) in the Carpathian Basin. North-Western Journal of Zoology. 2020 Dec 1;16(2).

Schofield, H., Reiter, G., Dool, S.E. (2022). Lesser Horseshoe Bat Rhinolophus hipposideros (André, 1797). In: Hackländer, K., Zachos, F.E. (eds) Handbook of the Mammals of Europe. Handbook of the Mammals of Europe. Springer, Cham. https://doi.org/10.1007/978-3-319-65038-8_39-1

SWARMING BEHAVIOUR

A very short window of time (only the end of August) is not enough to declare some sites as swarming sites. A longer time window should be considered for sampling, and also to trap during the whole night, as some species swarm later in the night than others (Parsons et al 2003).

Only 60 bats were captured during trapping efforts, which is also too small number of individuals to bring conclusions about sex ratio and if some site is a swarming site.

A little is known about the swarming phenomenon, but as the authors of this ms stated themselves, some of the functions of swarming sites are supposed to be mating, exchange of information etc. In the case of the first, the presence of both sexes of adult bats is expected. In case of later, the presence of adult females and youngs-of-the-year is expected (Burns & Broders 2015)

The authors conclude that, due to the extensive presence of display calls of Vespertilio murinus, the investigated area represents a swarming site. V. murinus is a highly synanthropic species that roosts in buildings, and roosting behaviour is not known for this species (Safi 2022). This is a migratory species, and it is very likely that recorded activity was from individuals on migration

References:

Parsons KN, Jones G, Greenaway F. Swarming activity of temperate zone microchiropteran bats: effects of season, time of night and weather conditions. Journal of Zoology. 2003 Nov;261(3):257-64.

Burns LE, Broders HG. Who swarms with whom? Group dynamics of Myotis bats during autumn swarming. Behavioral ecology. 2015 May 1;26(3):866-76.

Safi K. Parti-Colored Bat Vespertilio murinus Linnaeus, 1758. In: Handbook of the Mammals of Europe 2022 Mar 27 (pp. 1-12). Cham: Springer International Publishing.

 MINOR COMMENTS

Line 56: cryotograms instead of cryptograms

Line 126: remove 08 between of..their

Line 270 (Figure 4): why sex ration is shown only for one site? Again, sample size is too small to draw such conclusion
Line 314: Rydel instead of Reydel

Table 3: daubentonii instead of daubentoniid

Comments on the Quality of English Language

I am not a native speaker, but it seems that the quality of English language in this manuscript requires quite a lot of polishing. 

Author Response

Rev 3

In this manuscript, the authors investigate bat diversity at high elevations in Bulgaria using various methods. This topic is very interesting, but I am concerned about analysis methods and interpretation of their results.

I will provide some comments along with references that might help improving the quality of the manuscript.

DISCREPANCIES

From the sentence in the abstract (line 14) it is not clear that in all three years sampling was performed only at the end of August. In the abstract authors refer to data collected in 2013 and 2019-2020, and there is skeletal material included that was collected in 2002 (lines 116-117).

Response: We agree with reviewer comments and provide the require corrections (line 15).

Lines 22-23 authors state „Within all species adult males and juveniles predominated.“ while in lines 259-260 they state „all the animals captured were juveniles“ (contradiction between all juveniles and adults+juveniles)

Response: We agree with reviewer comments and provide the require corrections (line 23 and 272).

Lines 121-122: Please provide a reference to support this statement (that bats remained blocked due to snow)

Response: Lines 58 - 59 of the introduction provide information on this matter: „Some of the limestone caverns are characterised by the presence of ice and snow plugs that do not melt every summer. “

Lines 219, 369 why do authors use Myotis aurascens? Based on Çoraman et al 2020 and EUROBATS resolution 8.2, M. aurascens (M. mystacinus bulgaricus) is synonym of M. davidii

Response: We agree with reviewer comments and provide the require corrections (line 231).

Lines 431-433 (conclusios): It cannot be concluded that high parts of the Pirin Mountains provide suitable summer roosts without actual sampling during summer period and confirming reproduction (=existence of maternity roosts) there. Bats that were born that year are maybe just dispersing, it doesnt mean that they were born in that exact area

Response: We agree with reviewer comments and provide the require corrections (line 462-464).

References:

Çoraman E, Dundarova H, Dietz C, Mayer F. Patterns of mtDNA introgression suggest population replacement in Palaearctic whiskered bat species. Royal Society Open Science. 2020 Jun 3;7(6):191805.

https://www.eurobats.org/sites/default/files/documents/pdf/Meeting_of_Parties/MoP8.Resolution%208.2%20Amendment%20of%20the%20Annex%20to%20the%20Agreement_0.pdf

ACOUSTIC IDENTIFICATION

Authors should be more cautious and conservative with acoustic identification. It is true that bat surveys in such habitats are difficult, and that acoustics represents a great tool for fauna surveys, but a more robust approach should be considered. It is unclear how recordings were made – was the device set up for continuous recording, or only when there was a bat flying? In case of later, was the recording ongoing as long as bat echolocation calls were detected?

Response: The device was set up for continuous recording.

Could authors justify why they chose to cut their recordings in 1s? There are some papers suggesting that 5s files are of good length to compromise the detection distance (Millon et al 2015). Did authors check noise files as well, to be sure there were no files rejected? I do not have experience with BatScope 4, but in my experience, Kaleidoscope sometimes discards files with horseshoe echolocation calls as “noise” files.

Response: Regarding the selected length of 1 sec, a clarification has been added:Given the large volume of acoustic material, the length of 1 sec offers an optimal length in terms of subsequent manual verification, as it allows each sonogram to be fully opened on the computer screen, saving the need for scrolling and each call is well depicted with its "height" corresponding approximately to its length.“

The files classified as noise are not checked. Our experience with Kaleidoscope shows that weak signals are identified, i.e. not classified as noise, but not determined (noID). During the manual check in some cases, they are registered and taken into account - most often these were signals of Plecotus sp.

The next issue I would like to raise is identification to species level instead of sono-group for some „tricky“ species. There has been a lot of discussion about the unambiguous identification of certain species based on echolocation calls only (and in the absence of the social calls) such as species from genera Eptesicus / Nyctalus / Vespertilio. In many papers, authors group them into sono-group instead of identifying them to species level with lower certainty. In this paper, authors recorded display calls of V. murinus, which allows its reliable identification, but I would suggest authors to be a little more prudent with other species from the sono-group. For example, Rydell et al (2016) stress the very high percentage of erroneously identified recordings of species from this sono-group.

Response: We are aware of the difficulties in acoustic identification of species, therefore the identification procedure is presented in detail 172 – 204. For greater objectivity, the following text has been added: “However, realizing that due to the similarity of the signals and the wide overlap in their diagnostic characteristics, as well as the lack of other than acoustic evidence for the presence of these species in the area, an open nomenclature was used for these species, with the exception of Vespertilio murinus whose presence in the area is evidenced by the numerous social sounds (Table 2).”

The same authors point out frequent misclassifications of B. barbastellus. Authors in current ms present measurements of echolocation calls of this species. However, B. barbastella has two types of calls, and the authors did not specify which measurements (from which call type) were presented in their manuscript?

Response: In the verification process, cases of incorrect automatic determinations such as B. barbastellus, which were actually weak signals of Myotis, were recorded and corrected based on the gradual change in signal characteristics of what was undoubtedly a single individual within a single sonogram. For greater objectivity, sonograms illustrating the determinations made were given in the appendix. Also given was a table of dimensions automatically generated by the Kaleidoscope program. However, at the suggestion of one of the reviewers, in the process of the current revision of the manuscript, the table of measurements and sonograms in the appendix have been removed. Instead, 179 sonograms have been uploaded to a public database, where they are available for independent analysis and verification.

  1. hipposideros presence was confirmed based only on a single pass, and call measurements were not provided in Table 2. R. hipposideros echolocation calls are known to overlap with ones from R. euryale (Győrössy et al 2020, Schofield et al 2022)

Response: Added necessary clarification: „Rhinolophus hipposideros. The average value of the peak frequency of the available signals is 105.983 kHz (ChiroVox ID -AA004648, AA00464). This value, although close, is above the upper limit of overlap with Rh. euryale and is close to the average value of Rh. hipposideros (Győrössy et al., 2020) “.

 References:

Millon L, Julien JF, Julliard R, Kerbiriou C. Bat activity in intensively farmed landscapes with wind turbines and offset measures. Ecological Engineering. 2015 Feb 1;75:250-7.

Rydell J, Nyman S, Eklöf J, Jones G, Russo D. Testing the performances of automated identification of bat echolocation calls: A request for prudence. Ecological indicators. 2017 Jul 1;78:416-20.

Győrössy D, Győrössy K, Estok P. Comparative analysis of the echolocation calls of the lesser horseshoe bat (Rhinolophus hipposideros) and the Mediterranean horseshoe bat (Rhinolophus euryale) in the Carpathian Basin. North-Western Journal of Zoology. 2020 Dec 1;16(2).

Schofield, H., Reiter, G., Dool, S.E. (2022). Lesser Horseshoe Bat Rhinolophus hipposideros (André, 1797). In: Hackländer, K., Zachos, F.E. (eds) Handbook of the Mammals of Europe. Handbook of the Mammals of Europe. Springer, Cham. https://doi.org/10.1007/978-3-319-65038-8_39-1

SWARMING BEHAVIOUR

A very short window of time (only the end of August) is not enough to declare some sites as swarming sites. A longer time window should be considered for sampling, and also to trap during the whole night, as some species swarm later in the night than others (Parsons et al 2003).

Only 60 bats were captured during trapping efforts, which is also too small number of individuals to bring conclusions about sex ratio and if some site is a swarming site.

Response: We agree with reviewer comments. However, the following study was provided in a high mountain altitude which requires good weather conditions for claiming to rich the sites. In addition, the research was provided without funding and the authors chose was to go there in the most convenient period for bat research. However, it was not possible to capture bats all night long because of the constant stone fall.

A little is known about the swarming phenomenon, but as the authors of this ms stated themselves, some of the functions of swarming sites are supposed to be mating, exchange of information etc. In the case of the first, the presence of both sexes of adult bats is expected. In case of later, the presence of adult females and youngs-of-the-year is expected (Burns & Broders 2015)

Response: We agree with reviewer comments. However, most of the research was conducted during the bat peak of activity around midnight with the arriving of the first individuals until the decrease until 4am (Piksa 2008, Furmankiewicz 2008). In addition, we captured subadults which could be a sigh for both statements (line 429).

  • Piksa, K. Swarming of Myotis mystacinus and other bat species at high elevation in the Tatra Mountains, southern Poland. 2008. Acta Chiropterologica, 10, 69–79 (line 640)
  • Furmankiewicz, J. 2008. Population size, catchment area, and sex-influenced differences in autumn and spring swarming of the brown long-eared bat (Plecotus auritus). Canadian Journal of Zoology, 86(3), 207-216. (line 634)

The authors conclude that, due to the extensive presence of display calls of Vespertilio murinus, the investigated area represents a swarming site. V. murinus is a highly synanthropic species that roosts in buildings, and roosting behaviour is not known for this species (Safi 2022). This is a migratory species, and it is very likely that recorded activity was from individuals on migration

Response: We agree with reviewer comments (line 417-419). However, we cannot completely exclude that it correlated with the swarming because another hypothesis is that bats may use swarming sites as transitional stops during migration (Furmankiewicz 2008; Fenton 1969).

  • Furmankiewicz, J. 2008. Population size, catchment area, and sex-influenced differences in autumn and spring swarming of the brown long-eared bat (Plecotus auritus). Canadian Journal of Zoology, 86(3), 207-216. https://doi.org/10.1139/Z07-134
  • Fenton, M. B. 1969. Summer activity of Myotis lucifugus (Chiroptera:Vespertilionidae) at hibernacula in Ontario and Quebec. Canadian Journal of Zoology, 47(4), 597-602. https://doi.org/10.1139/z69-103

References:

Parsons KN, Jones G, Greenaway F. Swarming activity of temperate zone microchiropteran bats: effects of season, time of night and weather conditions. Journal of Zoology. 2003 Nov;261(3):257-64.

Burns LE, Broders HG. Who swarms with whom? Group dynamics of Myotis bats during autumn swarming. Behavioral ecology. 2015 May 1;26(3):866-76.

Safi K. Parti-Colored Bat Vespertilio murinus Linnaeus, 1758. In: Handbook of the Mammals of Europe 2022 Mar 27 (pp. 1-12). Cham: Springer International Publishing.

 MINOR COMMENTS

Line 56: cryotograms instead of cryptograms

Response: We agree with reviewer comments and provide the require corrections (line 57).

Line 126: remove 08 between of..their

Response: We agree with reviewer comments and provide the require corrections (line 129).

Line 270 (Figure 4): why sex ration is shown only for one site? Again, sample size is too small to draw such conclusion

Response: Because it is the only convent site for such research in the area.

Line 314: Rydel instead of Reydel

Response: We agree with reviewer comments and provide the require corrections (line 332).

Table 3: daubentonii instead of daubentoniid

Response: We agree with reviewer comments and provide the require corrections (line 15).

I am not a native speaker, but it seems that the quality of English language in this manuscript requires quite a lot of polishing.

Response: The linguistic proofreading of the manuscript was provided by a native speaker (line 448).

Reviewer 3 Report

Comments and Suggestions for Authors

This is a well-written manuscript on a moderately important topic of bat faunistic survey.

It is always hard to judge the soundness of the results of acoustic surveys. There is often an overlap in the acoustic parameters of different species and they cannot be identified to species level. The authors interpreted these results with caution, but in such cases, the reviewers cannot decide whether they are correct or not.

It would be good to deposit at least a part of the echolocation recordings in public acoustic libraries, like ChiroVox. In this case, the calls could be cited by their unique identifier and researchers could later verify the identifications.

I recommend to use more careful wording in connection with the swarming hypothesis, because most species were mist-netted in very low numbers (at normal swarming sites hundred of bats from the same species occur during a single night).

There were a few very minor type errors which I indicated in the attached manuscript.

Author Response

Response: We agree with reviewer comments and provide the required corrections. We are also grateful for the valuable advice.

Round 2

Reviewer 2 Report

Comments and Suggestions for Authors

Authors responded to the minor comments, but very little change was made regarding major comments. 

For example, for my comment:
"Lines 431-433 (conclusios): It cannot be concluded that high parts of the Pirin Mountains provide suitable summer roosts without actual sampling during summer period and confirming reproduction (=existence of maternity roosts) there. Bats that were born that year are maybe just dispersing, it doesnt mean that they were born in that exact area" authors responded "
We agree with reviewer comments and provide the require corrections (line 462-464)." And I cannot see any corrections in lines mentioned above. 

None of my comments about the swarming behaviour was addressed.
It is understandable how inaccessible investigated areas are, and how much hard work must be put in to collect such data, and no one is underestimating efforts in data collection. The data collected is new and valuable, but I disagree with conclusions about sex ratio and swarming behaviour., because time frame (both in dates - only several days in August; and in time - from 20h until midnight) is not enough for such conclusions.
I suggest another reviewer with expertise on the topic to provide comments 

Comments on the Quality of English Language

/

Author Response

Response: We clarified the conclusions as much as we could (line 465-468). Considering the swarming behaviour there are several hypotheses which we include in line 430-431. We believe that Site 3: Cave 30 is a swarming site because several forest-dwelling species (Myotis bechsteinii, M. mystacinus, M. nattereri) were present with peak of activity around midnight and decrease until 4am. These bat species are registered in the lower altitude but unfortunately, all records stay unpublished due to conflict of interests of some parties. For sure further investigation has to be provided including a vertical migration from the forest to the alpine zone. 

The authors collective is grateful for the reviewer comments, which helped us to improve our manuscript.